# Trust as a Determinant Factor for Condom Use among Female Sex Workers in Bali, Indonesia

**DOI:** 10.3390/tropicalmed5030131

**Published:** 2020-08-15

**Authors:** Pande Putu Januraga, Hailay Abrha Gesesew, Paul R. Ward

**Affiliations:** 1Center for Public Health Innovation, Faculty of Medicine, Udayana University, Bali 80232, Indonesia; januraga@unud.ac.id; 2College of Medicine and Public Health, Flinders University, Adelaide 5042, Australia; paul.ward@flinders.edu.au; 3Epidemiology, College of Health Sciences, Mekelle University, Mekelle 231, Ethiopia

**Keywords:** social capital, trust, social cohesion, HIV, inconsistent condom use, female sex workers, Indonesia

## Abstract

Female sex workers (FSWs) decision to use or not to use condoms depends on several issues, including the decision to trust their client or not, a matter given little attention in previous research. This paper explores the role of trust in consistent condom use among FSWs. We used a cross-sectional survey among FSWs in Bali, Indonesia. The outcome variable for this study was condom use, and independent variables included sociodemographic characteristics, psychosocial factors, social capital dimensions and HIV prevention practices. In total, 406 FSWs participated in the study. Of these, 48% of FSWs used condoms consistently with paying clients over the last month. The following FSWs were less likely to consistently use condoms with clients: FSWs who did not trust that their peer FSWs will use condoms (AOR = 9.3, 95% CI, 3.3–26.2), FSWs who did not feel valued by the people at their location (AOR = 3.9, 95% CI, 1.4–11.6), FSWs who did not graduate from primary or never went to school (AOR = 2.4, 95% CI, 1.03–5.6), and FSWs who have worked more than five years as FSWs (AOR = 5.8, 95% CI, 1.2–29.2). Our results highlight higher rates of inconsistent condom use related to lower trust and feelings of being valued between FSWs, identifying areas for policy and practice attention.

## 1. Introduction

### 1.1. Background 

Human immunodeficiency virus (HIV) is an important public health disease which incurs billions of dollars in healthcare costs [1] and infects and kills millions of people worldwide in general and low-and middle-income countries in particular [2]. With nearly 37 million HIV positive people globally and no vaccine or curative treatment available [2], HIV infection prevention remains pivotal to fighting the HIV/AIDS pandemic. Although the diseases affects the general population, most at-risk populations (MARPS) including injecting drug users (IDU), female sex workers (FSW) and men who have sex with men (MSM) are at a higher risk [3]. For example, Indonesia reported an HIV prevalence of 5–20% among FSWs where the disease was predominantly among brothel-based FSWs [4]. World Health Organization defines sex work as ‘the provision of sexual services for money or goods’ [5]. Evidence even showed that the prevalence of HIV among brothel-based FSWs in Bali had raised from 0.62% in 2000 to 20.2% in 2010 [4].

Fidelity (with an uninfected partner) and consistent condom use are two ways to prevent HIV infection for sexually active individuals. While the predominant mode of HIV transmission around the globe in general, and Indonesia in particular, is unprotected sexual intercourse, consistent condom use remains low [4]. In particular, literature shows that consistent condom use remains low among FSWs [4] due to several reasons including but not limited to: implied lack of trust or faithfulness, peer pressure, less fear of contracting HIV, the influence of tradition, alcohol and drug abuse, refusal of the partner, power and gender issues [6,7,8]. While trusting a steady partner is one of the main reasons for inconsistent condom use, the impact of this social capital such as trust, reciprocity, perceived safety and social harmony is not well investigated. In particular, the role of trust on condom use among FSWs in Indonesia has not been studied. As such, we focused on the following two research questions.
(a)What are dimensions of social capital (trust, reciprocity, membership of group and participation, safety and value of life) that associated with condom use with paying clients of FSWs in Bali?(b)After adjusting for sociodemographic (age, education, economic status, housing, household status, duration of being sex workers and duration of being sex workers in the current location) and psychosocial (self-efficacy, perceived vulnerability, and HIV knowledge) factors, are there still any associations between dimensions of social capital with condom use with paying clients of FSWs in Bali?

### 1.2. Theory of Trust

The concept of trust has been complex interplay and evolves around the following analytical elements, as quoted from Richard Convisor (1973) [9]: (i) what outcomes the person believes are possible in the situation, (ii) how the person evaluates each possible outcome (i.e., what payoff or profit he associates with each possible outcome), (iii) how the person believes other persons in the situation evaluate the possible outcomes (i.e., others’ perceived payoffs), (iv) what the person believes others’ decisions will be, and (5) what norms are operating in the situation. Trust has received increasing attention at an individual, societal or organizational levels and indeed in all activities of human beings; and area of interest in management, health, economics, sociology and psychology. Hence, trust is investigated for an individual’s (interpersonal trust), society (social trust), or organizations (institutional trust) [10,11]. Sociology has been vital for understanding the role of trust in the relationship between individuals and interaction with their environments i.e., community participation, political involvement, judicial system, and most importantly health service utilization [12]. Sociology also offers us a portion of theoretical frameworks which provide a lens to explore the role of trust in the health system. For example, Anthony Giddes and Niklas Luhmann, who made a significant contribution to the trust literature, recognized two forms of trust, namely institutional and interpersonal [13,14,15]. While institutional trust is a trust placed in the institution (e.g., health care institution) or system (e.g., medical system), interpersonal trust is regarded as trust which is being negotiated between individuals and as a learned personal trait [14]. For instance, from the interpersonal trust perspective, in a scenario of condom use and FSWs, FSWs’ decision to use or not to use condoms may depend on the decision to trust their client or not. As such, in the present study, we will explore the role of trust in consistent condom use among FSWs in Indonesia.

## 2. Materials and Methods

### 2.1. Study Design

The study used a cross-sectional survey design to answer the research questions mentioned above.

### 2.2. Population, Sample Size and Sampling Strategies

Participants for this study included FSWs in Bali. Given the circumstances and context of FSWs, the recruitment was expected to be challenging. As such, participants recruitment was conducted with the help of Yayasan Kerti Praja (YKP), a local NGO who work closely with the FSWs communities in Bali. The NGO provided the numbers of FSWs working in nine locations in Bali. We then recruited a proportional number of respondents in each location using a convenience sampling technique resulting in 406 respondents. With only 863 FSWs mapped during initial recruitment, we argued that 406 respondents was adequate to detect differences in the multivariate procedures [16].

### 2.3. Variables, Measurements, and Data Collection Process

Condom use was dichotomized as either consistent or inconsistent. Condom use was determined by combining responses to two questions regarding condom use with regular and new clients over the last month. Respondents who answered both questions by stating they always used condoms were coded “0”, consistent condom use, and the remainder were coded “1”, inconsistent condom use. Variables used in this study were developed from a qualitative study that has been previously published [17,18], which provided an instrument that was grounded in the views and language of the participants [19]. The survey instrument included sociodemographic characteristics (age, education, income, housing, duration of work in current location and duration of being a FSW), psychosocial factors (HIV knowledge, perceived vulnerability, self-efficacy), social capital dimensions (trust, perceived safety, value of life, sense of belonging, reciprocity, social harmony, social responsibility, connections with peers at house, neighbors connections, group membership and participation in groups) and HIV prevention practices (reported condom use). We also adopted relevant questions used by previous studies to measure dimensions of social capital, particularly studies among FSWs such as that by Samuels et al. (2006) [20]. Details of social capital dimensions measured in this study are presented in Table 1. Meanwhile, for questions related to HIV knowledge prevention practices, we adopted and modified the Behavioral Surveillance Survey (BSS) developed by Family Health International [21] that has been used in series of behavioral surveys in Indonesia. The benefits of modifying questions from previous studies enhances the content validity of measurements [22] and provides an opportunity for comparison to enhance the reliability of measurements [23]. Five trained enumerators approached the FSWs to collect data. Questionnaire were administered using face-to-face interviews in a one-to-one situation, and the data collection were employed in their locations.

### 2.4. Data Analysis

Double data entry was conducted using a freeware program called CSPro (US Census Bureau, Washington, DC, USA) and then exported into STATA 12 software for further analysis. With relatively huge numbers of variables collected in the survey, the analysis focused on variables comprising social capital, psychosocial, and sociodemographic factors.

Univariate analysis was used to describe the central tendencies of sociodemographic subgroups, psychosocial factors, dimensions of social capital and HIV prevention practices of the respondents. Bivariate logistic regression was used to screen for possible associations between dimensions of social capital, psychosocial, and sociodemographic factors with condom use. Variables that were significantly associated with condom use (*p* < 0.05) were entered into a multiple logistic regression model of relationships between social capital and consistent condom use. Further two multivariate logistic regression models were developed to test the association between dimensions of social capital and condom use after adjusting with selected demographic characteristics (age, education, number of clients per week, housing, time at complex, and time as FSW) and psychosocial correlates of HIV prevention practices. Model 1 (M1) included statistically significant social capital variables only, and model 2 (M2) included significant psychosocial and sociodemographic variables then used, to adjust the association between significant social capital dimensions with condom use.

### 2.5. Ethical Consideration

The study received ethical clearance from the Social and Behavioral Research Ethics Committee Flinders University in Adelaide, Australia (number 5913 SBREC) and the Institutional Review Board of Yayasan Kerthi Praja in Bali, Indonesia (number 040/YKP-IRB/2012). The recruitment team provided a letter of introduction and information sheet prior to the interviews, participants who responded to the questionnaire were considered as implying consent to participate in the study. Interview participants were offered payment of Rp.50,000 ($AUD5) in recognition of the time spent in interviews and to recognize their valuable contribution to this research. Participants also received a free condom on the day of the interview. The research procedures were conducted in ways that ensured anonymity and confidentiality.

## 3. Results

### 3.1. Demographic Characteristics

In Table 2 we present socio-demographic characteristics of participants, such as most women had low levels of education, with only 11% of respondents completing high school (year 12) or higher education. The majority of respondents (62%) lived outside the location and came to brothels only for work. Half of the women lived with partners or had partners who visited them regularly. When considering the duration of sex work in the current location, the findings indicated half the respondents were new to the sex work industry, with durations of less than one year. Lastly, related to economic status, half the women were able to save money, with only 8% stating they were in a bad economic condition.

### 3.2. Condom Use, and Social Capital Dimensions and Psychosocial Factors among FSWs

Of all study participants, 193 (47.5%) FSWs used condoms consistently and 213 (52.5%) used condoms inconsistently with paying clients over the last month. Table 3 describes the social capital dimensions and psychosocial factors for condom use. For example, the respondents had low levels of general and particular trust in their peers, with only 13.5% of respondents trusting “very much” or “quite2 that their peers would commit to condom use. Respondents had much higher levels of trust that their pimps would support sex workers to use condoms, indicating the strong competition between FSWs and showing that they had more trust in their pimps than in their peers.

Indicators for psychosocial factors show contradictory results. The majority of respondents (87%) perceived high self-efficacy for avoiding HIV infection, but only 22% had correct knowledge related to HIV prevention; while only 8% perceived they had no risk, 10% perceived very small, 42% perceived high and 39% perceived very high risk of HIV infection.

### 3.3. Factors Affecting Condom Use among FSWs

Table 4 is the final model of the multivariate analysis which showed that after adjusting for selected socio-demographic variables, social capital variables that were still significantly associated with condom use were trust peers related sexual transmitted infections (STIs) condition, trust peers to commit on condom use, and value of life, while after adjusting for sociodemographic variables friends’ connection was found insignificantly associated with condom use over the last month. The three significant social capital variables found in this model were bigger or compared to the model, not including sociodemographic factors. For example, FSWs who did not “trust peers that they will commit on condom use” at all were 11 times (AOR = 11.2, 95% CI, 4.04–31.3) more likely to use condom inconsistently than those who did “trust peers that they will commit on condom use” always.

The sociodemographic variables that were significantly associated with condom use over the last month were education level and duration of sex work in Bali. FSWs who did not graduate from primary or never went to school were two times (AOR = 2.4, 95% CI, 1.03–5.6) more likely to inconsistently use condoms than those who had secondary or higher education. In relation to duration of sex work in Bali, women that had been working for more than 5 years were significantly more likely to inconsistently use condoms with their paying clients over the last month.

## 4. Discussion

The present study discussed the role of trust on consistent condom use, one of the areas given less attention despite its significant public health importance. More than half (52%) of the FSWs participated in this study used condoms inconsistently with their paying clients over the last month. This is consistent with studies conducted elsewhere which reported from 45% in Singapore [26] to 68.6% in Cambodia [27]. The prevalence of inconsistent condom use in these studies is significant and this would expose FSWs and clients to HIV and other STIs [28]. For example, in Mexico, 6% of FSWs who used condoms inconsistently were found HIV positive, and 28% of them for other sexually transmitted infections [28,29].

The literature shows there are several reasons for such a prevalent inconsistent condom use by the FSWs. For example, Januraga et al. [17] described a number of factors for inconsistent condom use, but in particular, they argued that if clients became lovers or romantic partners, their relationships with the FSWs would become on a foundation of love, trust, respect, emotional and often material support, and FSWs faced difficulty to refuse unprotected sex. Yi et al. [27] reported that FSWs who used condoms inconsistently were more likely to have a higher consumption of alcohol and use contraceptives other than condoms. The present study showed that FSWs who did not trust that their peers will commit on condom use were nine times more likely to use condoms inconsistently than their counterparts. This could be due to prioritizing economic over health security; some clients may want to only have sex without condoms or may negotiate to pay less when they have sex with condoms, and the FSWs may accept unprotected sex in order not to lose money. Two studies by Januraga et al. [17,18] found that FSWs were offered “big money” for sex without condoms and most FSWs accepted the offer so as to be a financially successful migrant worker.

The present study also found that the odds of using condoms inconsistently were increased fourfold for FSWs who did not perceive a “value of life” compared to FSWs who had a higher perception of the value of life. Although no previous literature has been published on this matter, the finding might indicate the importance of healthy behaviours for an individuals’ social position within the social field where they worked or lived. Similarly, as supported by other studies [30], women who did not graduate from primary or who never went to school were significantly more likely to use condoms inconsistently than those who had secondary or higher education. This could imply their reduced capacity to understand the short- and long-term complications of sexually transmitted diseases as a result of sexual intercourse without condoms. In addition, it could relate to a short-term perceived need to earn “big money” via sex without condoms and in order to become a successful migrant in their own eyes. In relation to the duration of sex work in Bali, women who had worked there for more than five years were significantly more likely to have used condoms inconsistently with their paying clients over the last month. This might be related to the severe competition between FSWs in Bali, where FSWs who had been working there for a long time might be less able to compete with younger FSWs and thus negotiated condom use with their paying clients. In relation to this, the present study also found that FSWs who never talked to friends in the same house/brothel were 60% more likely to use condoms consistently than those who talked friends very much. This may be linked to the duration of stay in a way that those FSWs who stay for a short period of time might be newcomers and have few connections, which may prohibit them from knowing the existing competition among the “senior” FSWs in and around the house who perform sex without condoms to earn the “big money”.

The present study has a few limitations. First, the fact that the FSWs were recruited using convenient sampling technique may not be as representativeness as possible like those who would be recruited using simple random sampling or systematic random sampling technique. Nevertheless, 406 of the available 863 FSWs were included, and this significant proportion reduces the bias. Secondly, qualitative exploration of trust and [in] consistent condom use could give a detailed exploration of the mechanism of how trust impacts and the complex negotiation between social capital and condom use. Third, the FSWs were interviewed by a male interviewer and the primary author of the study, and this may bias the process of data collection towards the investigator (investigator bias) and gender bias during interviews. Future studies should consider this limitation. Nevertheless, the fact that the interviewer knows the local language, norm and concerns raised by the women in Bali and Indonesia could reduce these biases. Finally, this study only considered the nature of social factors among low-paid brothel-based FSWs in Bali. There are other types of FSWs that contribute to the broader complex picture of FSWs’ social-structural contexts that influence their condom use negotiations. Henceforth, more detailed qualitative and quantitative analysis is required to compare and examine comprehensively the interaction of social-structural factors within different groups of FSWs in Bali in relation to HIV prevention practices, particularly condom use negotiations. It is important for HIV prevention policy and practice in Bali to have answers to questions such as: do FSWs working in different sex work fields have different accounts of the nature of social and structural factors that shape their problems and the role of social capital in solving them?

## 5. Conclusions

In conclusion, more than half of the FSWs did not use condoms consistently. FSWs who used condoms inconsistently were those FSWs who: did not trust peers that they will commit on condom use, did not feel valued by the people at their location, did not graduate from primary/never went to school nearly, and worked as FSWs for more than 5 years. Work is required to increases levels of trust and mutual understanding between FSWs in order to increase consistent condom use and in so doing, reduce the transmission of HIV and other sexually transmitted infections.

## Figures and Tables

**Table 1 tropicalmed-05-00131-t001:** The measurement of social capital constructs.

	Dimensions of Social Capital	Proxy Indicators
Social Norm	Trust consisted of 4 options which were then coded as: Trust completely Trust somewhat Do not trust very much No trust at all	General trust
		Trust peers related to STIs * conditions
		Trust peers to share personal problems
		Trust peers to commit on condom use
		Trust pimp related to STIs condition
		Trust pimp to share problem
		Trust pimp to support condom use
		Trust pimp for equal treat
		Trust health personnel at STIs clinic
		Trust NGO ** staff
	Feeling of safety	feel safe at work
	Value of life	Feel valued by brothel’s community
	Sense of belonging	Perceived part as SWs community
	Reciprocity	Provided help
		Received help
	Social harmony	Perceived community problem
	Social responsibility	Would other SWs will provide helps when needed
Social network	Friends connections	Speak to peers from the same house
	Neighborhood connections	Speak to other SWs from other houses
	Participation in SWs group(s)	Group membership(s)
	Frequency of SWs actions/meetings/activities	Number of participations in the last six months

* STIs, sexual transmitted infections; ** NGO, non-governmental organizations.

**Table 2 tropicalmed-05-00131-t002:** Socio-demographic characteristics of direct FSWs.

Variable	%	(*n*)
**Age in years**		
18–24	21.92	(89)
25–29	21.43	(87)
30–34	24.14	(98)
35–39	19.70	(80)
≥40	12.81	(52)
**Education**		
Higher/secondary high	11.08	[24]
Secondary junior	25.86	(105)
Primary	37.19	(151)
Did not graduate from primary or never went to school	25.62	(104)
Missing	0.25	(1)
**Housing**		
In location	37.93	(154)
Outside location	62.07	(252)
**Household status**		
Live with husband or partner	35.96	(146)
Have a partner who visits regularly	14.78	(60)
Live alone	49.26	[25]
**Financial status**		
Saving money	50	(203)
Just getting by	41.63	(169)
Bad	8.37	(34)
**Duration of sex work in Bali**		
1 month	9.61	(39)
2–6 months	21.18	(86)
7–11 months	7.88	(32)
1–4 years	44.58	(181)
≥5 years	16.75	(68)
**Duration of sex work in current location**		
1 month	13.55	(55)
2–6 months	27.34	(111)
7–11 months	10.10	(41)
1–4 years	39.90	(162)
≥5 years	9.11	(37)
**District**		
Denpasar	76.60	(311)
Badung	23.40	(95)

**Table 3 tropicalmed-05-00131-t003:** Social capital dimensions, psychosocial factor, and consistent condom use of FSWs.

Variable	Yes, Very Much, *n* (%)	Quite, *n* (%)	Not Very, *n* (%)	Not at All, *n* (%)	Missing, *n* (%)
General trust *	135 (33.3)	135 (33.3)	235 (57.9)	34 (8.3)	2 (0.5)
Trust peers related STIs condition *	92 (22.7)	92 (22.7)	209 (51.4)	105 (25.9)	0
Trust peers for sharing problems *	68 (16.7)	68 (16.7)	233 (57.4)	105 (25.9)	0
Trust peers that they will commit on condom use *	55 (13.6)	55 (13.6)	275 (67.7)	71 (17.5)	5 (1.2)
Trust pimp related STIs condition	43 (10.6)	167 (41.1)	143 (35.2)	52 (12.8)	1 (0.3)
Trust pimp for sharing problem	41 (10.1)	140 (34.5)	154 (37.9)	71 (17.5)	0
Trust pimp that he/she will support SW to commit on condom use	64 (15.8)	152 (37.4)	158 (38.9)	32 (7.9)	0
Trust pimp for equal treat	24 (20.7)	191 (47)	98 (24.1)	30 (3.4)	3 (0.7)
Trust health personnel at STIs clinic **	130 (32)	208 (51.2)	67 (16.5)	67 (16.5)	1 (0.3)
Trust NGO staff **	115 (28.3)	212 (52.2)	79 (19.5)	79 (19.5)	0
Feeling of safety **	49 (12.1)	292 (71.9)	65 (16)	65 (16)	0
Value of life **	42 (10.3)	291 (71.7)	73 (17.9)	73 (17.9)	0
Sense of belonging **	71 (17.5)	218 (53.7)	117 (28.8)	117 (28.8)	0
Perceived social responsibility (peers will help when needed)	23 (5.7)	258 (63.5)	86 (21.2)	86 (21.2)	25 (1.7)
Friends connections	170 (41.9)	166 (40.9)	17.2	0	0
Neighborhood connections	0	30 (7.4)	174 (42.9)	202 (49.8)	0

* The same number of participants responded “Yes, very much” and “Quite”; ** The same number of participants responded “Not very” and “Not at all”; NGO, non-governmental organization; STI, sexually transmitted infections.

**Table 4 tropicalmed-05-00131-t004:** Multivariate analysis of social capital and condom use.

Variables	M1 Odds Ratio (95% CI)	M2 Odds Ratio (95% CI)
**Trust peers related STIs condition**		
Yes, very much/quite	1	1
Not very	0.7 (0.4–1.3)	0.8 (0.4–1.5)
Not at all	0.3 (0.1–0.6) **	0.4 (0.2–0.8) ***
**Trust peers that they will commit on condom use**		
Yes, very much/quite	1	1
Not very	3.3 (1.5–7.2) **	4.6 (2.1–10.3) *
Not at all	9.3 (3.3–26.2) *	11.2 (4.04–31.3) *
**Trust pimp related STIs condition**		
Yes, very much	1	
Quite	1.08 (0.4–2.8)	
Not very	0.9 (0.3–2.7)	
Not at all	2.6 (0.8–8.9)	
**Trust pimp that he/she will support SW to commit on condom use**		
Yes, very much	1	
Quite	1.3 (0.5–3.4)	
Not very	2.5 (0.8–7.6)	
Not at all	1.8 (0.5–7.2)	
**Trust pimp for equal treatment**		
Yes, very much	1	
Quite	0.9 (0.3–2.3)	
Not very	1.05 (0.3–3.3)	
Not at all	0.5 (0.1–1.9)	
**Feeling of safety (feel safe at work)**		
Yes, very much	1	
Quite	1.05 (0.5–2.4)	
Not very /not at all	1.3 (0.5–3.5)	
**Value of life (feel valued by people at location)**		
Yes, very much	1	1
Quite	2.2 (0.9–5.5)	2.2 (0.9–5.3)
Not very/not at all	3.9 (1.4–11.6) ***	4.4 (1.6–12.2) **
**Social harmony (perceived community problems)**		
No problem at all	1	
Not big	1.7 (0.9–3.4)	
Fairly big/very big	2.1 (0.9–5.2)	
**Friends connections (speak to peers from the same house)**		
Yes, very much	1	1
Quite	1.09 (0.6–1.9)	1.4 (0.8–2.4)
Not very/not at all	0.4 (0.2–0.8) **	0.6 (0.3–1.2)
**Age groups**		
18–24		1
25–29		1.4 (0.7–2)
30–34		1.4 (0.7–2.8)
35–39		1.7 (0.8–3.6)
≥40		1.9 (0.8–4.8)
**Education**		
Higher/secondary high		1
Secondary junior		1.7 (0.8–3.8)
Primary		1.6 (0.7–3.4)
Didn’t graduate from primary/never went to school		2.4 (1.03–5.6) ***
**Housing**		
Location		1
Outside location		1.4 (0.9–2.3)
**Duration of sex work in Bali**		
1 month		1
2–6 months		2.3 (0.4–12.06)
7–11 months		2.8 (0.5–16.6)
1–4 years		2.8 (0.7–11.9)
≥5 years		5.8 (1.2–29.2) ***
**Duration of sex work in current location**		
1 month		1
2–6 months		1.3 (0.3–5.2)
7–11 months		0.6 (0.1–2.9)
1–4 years		0.8 (0.2–2.6)
≥5 years		0.4 (0.09–2.08)
**Districts of brothels**		
Denpasar		1
Badung		1.6 (0.9–2.8)

* *p* <0.001, ** *p* <0.01, *** *p* <0.05; STIs, sexual transmitted infections.

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
