# Peer review of "Trust as a Determinant Factor for Condom Use among Female Sex Workers in Bali, Indonesia"

_tropicalmed, 2020, doi:10.3390/tropicalmed5030131_

Round 1

Reviewer 1 Report

I have not conducted research on this topic but I am interest in trust and its importance in shaping relationships. My comments should be taken as someone interested in the topic but not an "expert" on the topic.

 The authors do not probe the psychological dimensions in their broad base survey. The study does benefit, however, with having a native speaker participate in the interviews which enhances the overall quality or reliability. 

It is well known that trust and love are intertwined you cannot love someone if you do not trust them. This has been one of the primary reasons why Thai wife's feel they cannot insist their husband wear a condom when he insists he has not been with another woman. If she loves him, she cannot insist. In the Bali case we see long term sex workers are caught in a similar and not so similar bind and the desire for and the hope for a more positive outcome (HIV free)  influences their judgment. In early 2000s Chinese men sought extramarital sexual relationships with "girlfriend" and not prostitute as they thought they could trust them to be STD free. Hope and trust can flow both ways.

 In this fine survey I think the authors have expanded upon the role of trust and its significance for understanding  the preference for and against the use of condoms. 

Reviewer 2 Report

OVERVIEW

Thanks for this interesting paper, which looks at an important topic in the lives of female sex workers in Indonesia. The paper is generally well-written, though there are a few typos that will need correcting.

BACKGROUND

This provides a useful background, cites recent literature, and provides suitable justification for the study. I would recommend that you provide a clearer definition of female sex worker. Also, on line 56, please remove ‘and AIDS’.

I appreciate you flag something of the complexities in the discussion, and also you do specify you respondents are brothel-based. But, as I’m sure you’re aware, some sex workers work only a few days a month, others every night, and others once a month. Is this distinction relevant to your study?

The theory of trust is interesting, and the sociological perspective I think is important. Please note that the Richard Convisor citation needs to be included in the reference list at the end of the document.

METHODS

This includes useful detail of the approach taken to data collection and analysis. The design seems appropriate, and uses existing tools adopted for this study. You state that data were collected through face-to-face interactions.

By ‘their place of work’, do you mean in the brothel itself? How difficult was it to find somewhere private? This would be useful for readers to know (I’ve done similar research and it can be challenging).

I note that the study was given ethical approval and had a consenting process in place.

FINDINGS

This is a succinct section which provides helpful details of the findings. Table 4 is especially interesting – I would suggest adding a short sentence describing the significance of the data (the actual numbers) displayed, for readers who may not be familiar with multivariate analysis. The finding that women that had who been working for more than 5 years being significantly more likely to use condoms inconsistently is very intriguing (one would expect it to be the other way around).

DISCUSSION/CONCLUSION

This draws appropriately on the findings and discussion. You cite other sources to add depth and context, and this is appreciated. Your key finding – that trust in peers’ commitment to using condoms, and the likely financial reasons for this – is important to the field, as well as the ‘value of life’ variable. In this section, I would also recommend discussing the impact of the sex worker's age (I think you address most of the other variables).

I note that you include limitations and recommendations for further research, and these are appropriate. I also note that a limitation is a male data collector. I would certainly recommend in future that data are collected by a female interviewer if possible, especially for more qualitative studies. I appreciate in this case familiarity with the topic and issues are important, but it’s important to acknowledge gender dynamics and how this may influence responses in this case.

In the conclusion, I recommend amending the final sentence to end: 'in order to increase consistent condom use and in so doing, reduce the transmission of HIV and other sexually transmitted infections'. 

Reviewer recommendations

  1. Address typos throughout
  2. In the background section, add citation for Convisor, remove ‘and AIDS’, and add a clearer definition of ‘sex worker’
  3. In the methods section, add a little more detail about the logistics of data collection [which in this case can present additional difficulties given the context]
  4. In the findings section, add a short definition of the figures presented in Table 4
  5. In the discussion section, I would recommend adding a short sentence or two about the ‘age’ variable
  6. Acknowledge that for future research, further measures will be taken to ensure a gender match for data collection
  7. Amend last sentence (see comment)

Author Response

We have attached a point-by-point responses for both reviewers.

Thanks.
